# Simple Spectral Graph Convolution from an Optimization Perspective

## Abstract

Recent studies on SGC, PageRank and $S^2$GC have demonstrated that several graph diffusion techniques are straightforward, quick, and effective for tasks in the graph domain like node classification. Even though these techniques do not even need labels, they can nevertheless produce more discriminating features than raw attributes for downstream tasks with different classifiers. These methods are data-independent and thus primarily rely on some empirical parameters on polynomial bases (e.g., Monomial and Chebyshev), which ignore the homophily of graphs and the attribute distribution. They are more insensitive to heterophilous graphs due to the low-pass filtering. Although there are many approaches focusing on GNNs based on heterophilous graphs, these approaches are dependent on label information to learn model parameters. In this paper, we study the question: are labels a necessity for GNNs with heterophilous graphs? Based on this question, we propose a framework of self-representation on graphs related to the Least Squares problem. Specifically, we use Generalized Minimum RESidual (GMRES) method, which finds the least squares solution over Krylov subspaces. In theoretical analysis, without label information, we enjoy better features with graph convolution. The proposed method, like previous data-independent methods, is not a deep model and is, therefore, quick, scalable, and simple. We also show performance guarantees for models on real and synthetic data. On a benchmark of real-world datasets, empirically, our method is competitive with existing deep models for node classification.

## 1 Introduction

With the development of deep learning, CNNs have been widely used in different applications. A convolutional neural network (CNN) is exploits the shift-invariance, local connectivity, and compositionality of image data. As a result, CNNs extract meaningful local features for various image-related problems. Although CNNs effectively capture hidden patterns on the Euclidean grid, there is an increasing number of applications where data is represented in the form of non-Euclidean grid, e.g. in the graph domain.

GNNs redefine the convolution on the graph in two different ways: spatial and spectral. Spatial-based methods decompose the convolution operation into an aggregation function and a transformation function. The aggregation function is used to aggregate neighbourhood node information by the mean function, which is somewhat similar to the box filter in traditional image processing. Some representative methods in this category are Message Passing Neural Networks (MPNN (Gilmer et al., 2017)), GraphSAGE (Hamilton et al., 2017), GAT (Veličković et al., 2017), etc. Spectral methods are based on Graph Fourier Transformation (GFT). They try to learn a filtering function on the eigenvalues (or graph kernel, heat kernel, etc.) These methods usually use approximations in order to simplify the amount of computation, e.g. Chebyshev polynomials and Monomial polynomials are used by ChebNet (Defferrard et al., 2016)), GDC (Klicpera et al., 2019), SGC (Wu et al., 2019), $S^2$GC (Zhu & Koniusz, 2021). Although spatial and spectral methods effectively extend the convolution operator to the graph domain, they usually suffer from oversmoothing on heterophily graph because they follow the homophily assumption, thus severely affect the node classification task as shown in Figure 1.

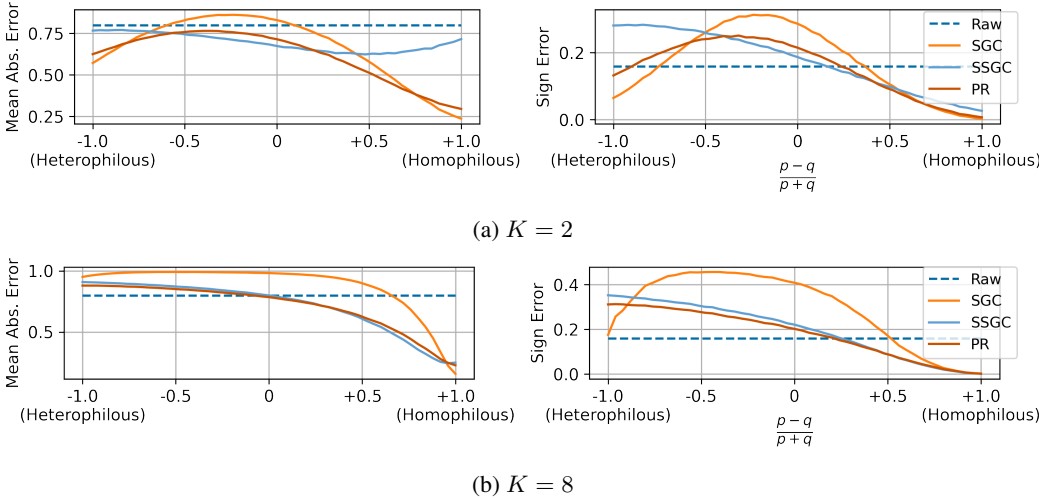

Figure 1: Results on the contextual SBM using SGC, S$^2$GC and PR (PageRank) with number of hops $K = 2, 8$. 'Raw' shows the error when no filtering method is applied. All methods only work well in homophilous networks.

However, graphs are not always homophilic: they show the opposite property in some connected node groups. This makes it harder for existing homophilic GNNs to learn from general graph-structured data, which leads to a significant drop in performance on heterophilous graphs. There are many GNNs for a graph with heterophily. Their motivation mainly focuses on improving feature propagation and features transformation. Non-local neighbor extension usually is used for incorporating high-order neighbor information (Abu-El-Haija et al., 2019; Zhu et al., 2020; Jin et al., 2021) or discovering potential neighbours (Liu et al., 2021; Yang et al., 2021; Zheng et al., 2022). Adaptive message aggregation is a good way to reduce the heterophilous edges (Veličković et al., 2017; Suresh et al., 2021). Inter-layer combination provide a more flexible way to learn graph convolution (Xu et al., 2018; Zhu et al., 2020; Chien et al., 2021). However, all of these approaches are designed for semi-supervised node classification, which is usually transductive (labels for training).

In this paper, first we review the connection between GNNs and the Label Propagation (LP) with Laplacian regularization (Zhou et al., 2003). The closed-form solution only depends on a parameter balancing smoothing and fitting error. This results in low-pass filter methods for homophilious graphs such as PageRank and S$^2$GC, which cannot work well on heterophilous graphs. Based on the Taylor expansion of the closed form solution, we reformulate label propagation with Laplacian regularization to Residual Minimization in Krylov subspace. We further generalize the residual minimization in the Krylov subspace into a more generalized Polynomial Approximation. Then we discuss other possible bases such as Chebshev polynomials. In theoretical analysis, we try to explore whether high-order (second-order in this paper) or multi-scale graph convolutions are able to improve the performance given raw attributes without labels. In experiments with synthetic data, we show performance in line with our theoretical expectations. On the real-world benchmarks, our method is competitive with other graph convolution techniques in homophilous graphs and outperforms them (even some GNNs methods with transductive learning) on heterophilous graphs.

Our contributions are: 1.) We reveal the labels are not necessary for graph neural networks on heterophilous graphs. The linear graph convolution is powerful on heterophilous graphs and homophilous graph, and outperforms GNNs for heterophilous graphs on semi-supervised node classification. 2.) We propose a framework of Feature (or Label) Propagation by parameterizing spectral graph convolution as residual minimization in Krylov subspace. We further reformulate residual minimization problem into Polynomimial Approximation, which can yield Chebshev and Berstein bases to overcome the Runge phenomenon. 3.) In theory, we prove second-order graph convolution is better than first-order graph convolution on heterophilous graphs and multi-scale (single and second-order) can provide better results with some combinations of parameters. 4.) Compared with other methods of label-dependent GNNs under heterophily, our method is competitive in real-world benchmarks. The proposed method outperforms other low-pass graph convolution without learning.

## 2 RELATED WORKS

**Data-independent Spectral Graph Convolution**  Hammond et al. (2011) introduced Chebyshev polynomial to estimate wavelets in graph signal processing. Based on this polynomial approximation, ChebNet was proposed for combining neural network and the graph convolution operator. Unlike ChebNet using Chebyshev polynomial, Diffusion Convolutional Neural Network (DCNN (Atwood & Towsley, 2016)) use the normalized adjacent matrix as polynomial bases to approximate any graph filters. Simplifying Graph Convolution (SGC (Wu et al., 2019)) are a special case of DCNN: only k-th power of normalized adjacent matrix is selected. Graph Diffusion Convolution (GDC (Klicpera et al., 2019)) shown two other special cases based on normalized adjacent matrix: heat kernel and PageRank kernel. It should be noted that GDC re-normalizes the given kernel like a normalized adjacent matrix. Thus, in this paper, we use PageRank kernel such as APPNP (Klicpera et al., 2018). Simple Spectral Graph Convolution (S$^2$GC (Zhu & Koniusz, 2021)) is based on a modified Markov diffusion kernel (Fouss et al., 2012). Although these methods are effective on node classification, fixed parameters ignore the graph property of homophily/heterophily, and node attributes in different dimensions. These drawbacks limit such methods on heterophilous graphs.

**Learnable graph convolutions**  Chebyshev polynomials are used by ChebNet (Defferrard et al., 2016) to approximate the graph convolutions. In theory, one can learn any kind of filter (Balcilar et al., 2021). With Cayley polynomials, CayleyNet (Levie et al., 2018) learns the graph convolutions and produces a variety of graph filters. Low-pass or high-pass filters can be derived from graph convolutions. GPR-GNN (Chien et al., 2021) employs the Monomial basis to approximate these filters. Through the family of Auto-Regressive Moving Average filters (Narang et al., 2013), ARMA (Bianchi et al., 2021) learns the rational graph convolutions. The graph convolutions are approximated in BernNet (He et al., 2021), which also learns graph filter using the Bernstein basis. Although these methods achieve good performance on different datasets, the learnable parameters of the graph convolution kernel depend only on the label information, which leads to overfitting due to too few or unbalanced labels.

**Graph Neural Networks for Heretophily**  Graphs are not always homophilic. The opposite is true on connected node groups. This makes it harder for existing homophilic GNNs to learn from general graph-structured data, which leads to a significant drop in performance on heterophilous graphs. Increasing Homophilic Edges (HoE) and decreasing Heterophilic Edges (HeE) are two mainly ways to improve feature propagation. HoE refers to edges connecting two nodes of the same class while HeH means edges connecting two nodes of different classes. The strategies of increasing increasing HoE include using two-hop (or higher) neighbours and discovering new neighbours with feature similarity. Decreasing Heterophilic Edges (HeE) assigns the weights on edges to reduce the impact from potential heterophilous edges. At each message passing step, H$^2$GCN (Zhu et al., 2020) aggregates data from higher-order neighbors. In order to offer theoretical guarantee, H$^2$GCN confirms that when one-hop neighbors' labels are conditionally independent, two-hop neighbors tend to include more nodes belonging to the same class. The generalised PageRank is used with graph convolutions in GPR-GNN (Chien et al., 2021) to jointly maximise the extraction of node features and topological information for both homophilous and heterophilous graphs. These methods are based on transductive learning. Without label information they cannot learn a useful model. Graph convolution based methods such as SGC, S$^2$GC and PageRank do not need labels at all.

## 3 METHODS

In this section, we review the classical Label Propagation with Laplacian Regularization (Zhou et al., 2003) and show the relationship between his iterative solution and the existing GNNs. Then, by analyzing the closed form of LP, we formulate the label propagation to residual minimizing in Krylov subspace to learn parameters for graph convolution. To overcome the Runge phenomenon, we reformulate residual minimizing problem in Krylov subspace to a more general polymonimial approximation problem, which helps introduce other kinds of bases such as Chebyshev and Berstein polynomials.

### 3.1 PRELIMINARIES

Let $G = (V, E)$ be a simple and connected undirected graph with $n$ nodes and $m$ edges. We use $\{1, \cdots, n\}$ to denote the node index of $G$, whereas $d_j$ denotes the degree of node $j$ in $G$. Let

$\boldsymbol{A}$ be the adjacency matrix and $\boldsymbol{D}$ be the diagonal degree matrix. Let $\widetilde{\boldsymbol{A}} = \boldsymbol{A} + \boldsymbol{I}_n$ denote the adjacency matrix with added self-loops and the corresponding diagonal degree matrix $\widetilde{\boldsymbol{D}}$, where $\boldsymbol{I}_n \in \mathbb{R}^{n \times n}$ is an identity matrix. Finally, let $\boldsymbol{X} \in \mathbb{R}^{n \times d}$ denote the node feature matrix, where each node $v$ is associated with a $d$-dimensional feature vector $\boldsymbol{x}_v$. To facilitate the definition of dimension-independent objective functions, we use $\boldsymbol{y} \in \mathbb{R}^{n \times 1}$ to denote 1D node features.

**Label Propagation with Laplacian Regularization.** A classical regularization framework for label (or feature) propagation (Zhou et al., 2003) includes two components: a fitting term with least square and a smoothing term with Laplacian regularization.The fitting term controls the target so it is not far away to the original point. The smoothing term encourages the connected elements have similar scale. The loss function associated with $\boldsymbol{f} \in \mathbb{R}^{n \times 1}$ is defined as:

$$\mathcal{E}(\boldsymbol{f}) = \frac{1}{2} \left( \sum_{i,j=1}^{n} A_{ij} \left\| \frac{1}{\sqrt{D_{ii}}} f_i - \frac{1}{\sqrt{D_{jj}}} f_j \right\|^2 + \mu \sum_{i=1}^{n} \| f_i - y_i \|^2 \right), \quad (1)$$

where $\mu > 0$ is the regularization parameter. Differentiating $\mathcal{E}(\boldsymbol{f})$ with respect to $\boldsymbol{f}$, we have

$$\left. \frac{\partial \boldsymbol{Q}}{\partial \boldsymbol{f}} \right|_{\boldsymbol{f}=\boldsymbol{f}^*} = \boldsymbol{f}^* - \boldsymbol{A}\boldsymbol{f}^* + \mu \left( \boldsymbol{f}^* - \boldsymbol{y} \right) = 0. \quad (2)$$

Although there exist a closed-form based on Eq. 2, for large graphs the inverse of $\boldsymbol{I} - \alpha\boldsymbol{A}$ is not practically-feasible to compute, and instead iterative approximations are preferable. To this end, we may set $\boldsymbol{f}^{(0)} = \boldsymbol{y}$, and then proceed to iteratively descend in the direction of the negative gradient:

$$\boldsymbol{f}^{(t+1)} = \alpha\boldsymbol{A}\boldsymbol{f}^{(t)} + (1-\alpha)\boldsymbol{f}^{(0)}, \quad (3)$$

where $\alpha = \frac{1}{1+\mu}$. If we define $\boldsymbol{y} = f(\boldsymbol{X}; \boldsymbol{\theta})$ and replace $\boldsymbol{A}$ with $\widetilde{\boldsymbol{A}}$, Eq. 3 equates to principled GNN layers, such as those used by GCN (Kipf & Welling, 2016), APPNP (Klicpera et al., 2018).

## 3.2 Label Propagation with Residual Minimizing over Krylov Subspace

In this section, we reformulate the closed-form solution of label propagation with Laplacian regularization (Zhou et al., 2003) to a more generalized model based on a Residual Minimizing over Krylov Subspace to solve for the parameters for graph convolution.

Let us remind the closed-form solution Eq. 1: $\boldsymbol{f} = (1-\alpha)(\boldsymbol{I} - \alpha\boldsymbol{A})^{-1}\boldsymbol{y}$. If we put the closed-form solution into the fitting term of Eq. 1, we have:

$$\min_{\boldsymbol{f}} \| \boldsymbol{y} - \boldsymbol{f} \|_2 = \min_{\alpha} \| \boldsymbol{y} - (1-\alpha)(\boldsymbol{I} - \alpha\boldsymbol{A})^{-1}\boldsymbol{y} \|_2 = \min_{\boldsymbol{w} \in \mathbb{R}^r} \| \alpha\boldsymbol{y} - \boldsymbol{A} \sum_{i=0}^{r} w_i \boldsymbol{A}^i \boldsymbol{y} \|_2, \quad (4)$$

where $w_i = (1-\alpha)\alpha^i$. We could rescale $\boldsymbol{y}$ by $1 - (1-\alpha)$ to eliminate the parameter $\beta$. Please note $r < \text{rank}(\boldsymbol{A})$. Then we obtain a more compact form:

$$\min_{\boldsymbol{w} \in \mathbb{R}^{r-1}} \| \boldsymbol{y} - \boldsymbol{A} \sum_{i=0}^{r} w_i \boldsymbol{A}^i \boldsymbol{y} \|_2 = \min_{\boldsymbol{x} \in \mathcal{K}_r(A,b)} \| \boldsymbol{y} - \boldsymbol{A}\boldsymbol{x} \|_2, \quad (5)$$

where the set of vectors $\mathcal{K}_r(\boldsymbol{A}, \boldsymbol{y}) = \left\{ \boldsymbol{y}, \boldsymbol{A}\boldsymbol{y}, \boldsymbol{A}^2\boldsymbol{y}, \dots, \boldsymbol{A}^{r-1}\boldsymbol{y} \right\}$ is called the order-$r$ Krylov matrix, and the subspace spanned by these vectors is called the order-$r$ Krylov subspace. Based on this we obtain a denoised signal as $\boldsymbol{f} = \boldsymbol{A}\boldsymbol{x}$.

**Relation to GPR-GNN (Chien et al., 2021).** GPR-GNN first extracts hidden state features with a MLP for each node and then uses Generalized PageRank (GPR) to propagate them. The GPR-GNN process can be mathematically described as:

$$\hat{\boldsymbol{P}} = \text{softmax}(\boldsymbol{Z}), \boldsymbol{Z} = \sum_{k=0}^{K} \gamma_k \boldsymbol{H}^{(k)}, \boldsymbol{H}^{(k)} = \tilde{\boldsymbol{A}}\boldsymbol{H}^{(k-1)}, \boldsymbol{H}^{(0)} = f\left(\boldsymbol{X}; \boldsymbol{\theta}\right), \quad (6)$$

where $\text{softmax}(\boldsymbol{Z}_{i,:}) = \frac{e^{Z_{ij}}}{\sum_{j=1}^{c} e^{Z_{ij}}}$. Although Generalized PageRank looks similar to the purpose of our approach, we notice three key differences: (1) The GPR learns generalized graph convolution on logits rather than features (or attributes). (2) The parameters in GPR only depend on labels rather than internal information of the graph and the corresponding attributes. (3) There is no global optimal solution for GPR-GNN because of the feature extraction with a MLP.

### 3.3 POLYNOMIAL APPROXIMATION WITH CONSTRAINTS

The Eq. 5 also solves an approximation problem. The only difference being that the space of polynomials is now $P_r = \{$ polynomials $p$ of degree $\leq r$ with $p_r(0) = 1\}$. Expressed in terms of polynomial coefficients, we have a constraint of $w_0 = 1$. Here is how Eq. 5 can be reduced to the polynomial approximation in $P_r$. The iterate $\boldsymbol{x}$ can be written as $\boldsymbol{x} = q_r(\boldsymbol{A})\boldsymbol{y}$, where $q_r$ is a polynomial of degree $r - 1$; its coefficients are the entries of the vector $\boldsymbol{w}$ of Eq. 5 . The corresponding residual $\boldsymbol{r} = \boldsymbol{y} - \boldsymbol{A}\boldsymbol{x}$ is $\boldsymbol{r} = (\boldsymbol{I} - \boldsymbol{A}q_r(\boldsymbol{A}))\boldsymbol{y}$. If we define $p_r(z) = 1 - zq_r(z)$, which is the polynomial, we have $\boldsymbol{r} = p_r(\boldsymbol{A})\boldsymbol{y}$ for some polynomial $p_r \in \mathcal{P}_r$. Thus, we can reformulate Eq. 5 as:

$$\min_{\boldsymbol{x} \in \mathcal{K}_r(A,b)} \|\boldsymbol{y} - \boldsymbol{A}\boldsymbol{x}\|_2 = \min_{p_r \in \mathcal{P}_r, p_r(0)=1} \|p_r(\boldsymbol{A})\boldsymbol{y}\|_2, \tag{7}$$

where $\mathcal{P}_r$ is the set of all polynomials $p_r$ of degree at most $r$ such that $p_r(0) = 1$.

Chebyshev polynomials are frequently employed in digital signal processing and graph signal filtering to approximate a variety of functions. The analytical functions may be approximated by a minimax polynomial using the truncated Chebyshev expansions. Consequently, a truncated expansion expressed in terms of Chebyshev polynomials can minimize the loss function as follows:

$$\min_{\boldsymbol{w} \in \mathbb{R}^r} \| \sum_{i=0}^{r-1} w_i T_i(\hat{\boldsymbol{L}})\boldsymbol{y}\|_2, \tag{8}$$

where $\hat{\boldsymbol{L}} = 2\boldsymbol{L}/\lambda_{\max} - \boldsymbol{I}$ denotes the scaled Laplacian matrix. $\lambda_{\max}$ is the largest eigenvalue of $\boldsymbol{L}$ and $w_k$ denote the Chebyshev coefficients. The Chebyshev polynomials can be recursively defined as $T_k(x) = 2xT_{k-1}(x) - T_{k-2}(x)$, with $T_0(x) = 1$ and $T_1(x) = x$. Although Chebyshev polynomials have many great properties such as relieving the Runge phenomenon, they underperform in GNNs. How to solve this problem is beyond the scope of this paper.

### 3.4 THEORETICAL ANALYSIS

We study our method in the contextual stochastic block model (cSBM) (Deshpande et al., 2018), which is a generative model for random graphs. For the purposes of theoretical analysis, we take into account a CSBM model with two classes, $c_0$ and $c_1$. The generated graphs in this instance have nodes made up of two distinct sets, $\mathcal{C}_0$ and $\mathcal{C}_1$, which represent the two classes, respectively. An intra-class probability $p$ and an inter-class probability $q$ are used to produce edges. In particular, an edge is constructed to connect any two nodes in the graph with probability $p$ if they belong to the same class, and $q$ otherwise. For each node $i$, its initial associated features $\boldsymbol{x}_i \in \mathbb{R}^l$ are sampled from a Gaussian distribution $\boldsymbol{x}_i \sim N(\boldsymbol{\mu}, \sigma\boldsymbol{I})$, where $\boldsymbol{\mu} = \boldsymbol{\mu}_k \in \mathbb{R}^l$ for $i \in \mathcal{C}_k$ with $k \in \{0, 1\}$. Hence, we denote a graph generated from such an cSBM model as $\mathcal{G} \sim \text{cSBM}(\boldsymbol{\mu}_1, \boldsymbol{\mu}_2, p, q)$, and the features for node $i$ obtained after a first-order graph convolution as $\boldsymbol{h}_i^1$ and $\boldsymbol{h}_i^2$ with second-order graph convolution.

Ma et al. (2022) propose a very interesting problem 'Is homophily a necessity for graph neural networks?' A very useful property has been proven that first-order graph convolution can provide a better features if the $deg(i) > \frac{(p+q)^2}{(p-q)^2}$ is met, which demonstrates that the node degree $deg(i)$ and the distinguishability (measured by the Euclidean distance) of the neighborhood distributions both affect graph convolution performance. This condition often happens in practice. Thus, we are interested whether or not the higher-order graph convolution still enjoy such a property. As the proposed method could be regarded as a multi-scale graph convolution, it is important to know whether there are existing parameters that make the multi-scale graph convolution better than single graph convolution.

To better evaluate the effectiveness of our method, we study the linear classifiers with the largest margin based on $\{\boldsymbol{x}_i, i \in \mathcal{V}\}$, $\{\boldsymbol{h}_i^1, i \in \mathcal{V}\}$ and $\{\boldsymbol{h}_i^2, i \in \mathcal{V}\}$ compare their performance. Here we define relation among $\boldsymbol{x}_i$, $\boldsymbol{h}_i^1$ and $\boldsymbol{h}_i^2$ as follows:

$$\boldsymbol{h}_i^1 = \frac{1}{\deg(i)} \sum_{j \in \mathcal{N}(i)} \boldsymbol{x}_j \text{ and } \boldsymbol{h}_i^2 = \frac{1}{\deg(i)} \sum_{j \in \mathcal{N}(i)} \boldsymbol{h}_j^1, \tag{9}$$

where $\mathcal{N}(i)$ denotes the neighbors of node $i$.

For a graph $\mathcal{G} \sim \mathrm{cSBM}\,(\boldsymbol{\mu}_1, \boldsymbol{\mu}_2, p, q)$, we can approximately regard that for each node $i$, its neighbor's labels are independently sampled from a neighborhood distribution $\mathcal{D}_{y_i}$, where $y_i$ denotes the label of node $i$. Specifically, the neighborhood distributions corresponding to $c_0$ and $c_1$ are $\mathcal{D}_{c_0} = \left[\frac{p}{p+q}, \frac{q}{p+q}\right]$ and $\mathcal{D}_{c_1} = \left[\frac{q}{p+q}, \frac{p}{p+q}\right]$, respectively. Based on the neighborhood distributions, the features obtained from Graph Convolution follow the Gaussian distributions:

$$\boldsymbol{h}_i^1 \sim N\left(\frac{p\boldsymbol{\mu}_0 + q\boldsymbol{\mu}_1}{p+q}, \frac{\boldsymbol{I}}{\sqrt{\deg(i)}}\right), \boldsymbol{h}_i^2 \sim N\left(\deg(i)(\frac{p^2\boldsymbol{\mu}_0 + 2pq\boldsymbol{\mu}_1 + q^2\boldsymbol{\mu}_1}{p+q}), \boldsymbol{I}\right), \text{ for } i \in \mathcal{C}_0,$$

$$\boldsymbol{h}_i^1 \sim N\left(\frac{q\boldsymbol{\mu}_0 + p\boldsymbol{\mu}_1}{p+q}, \frac{\boldsymbol{I}}{\sqrt{\deg(i)}}\right), \boldsymbol{h}_i^2 \sim N\left(\deg(i)(\frac{p^2\boldsymbol{\mu}_1 + 2pq\boldsymbol{\mu}_0 + q^2\boldsymbol{\mu}_0}{p+q}), \boldsymbol{I}\right), \text{ for } i \in \mathcal{C}_1. \tag{10}$$

**Proposition 1.** *(Ma et al., 2022) $(\mathbb{E}_{c_0}[\boldsymbol{x}_i], \mathbb{E}_{c_1}[\boldsymbol{x}_i])$ and $(\mathbb{E}_{c_0}[\boldsymbol{h}_i], \mathbb{E}_{c_1}[\boldsymbol{h}_i])$ share the same middle point. $\mathbb{E}_{c_0}[\boldsymbol{x}_i] - \mathbb{E}_{c_1}[\boldsymbol{x}_i]$ and $\mathbb{E}_{c_0}[\boldsymbol{h}_i] - \mathbb{E}_{c_1}[\boldsymbol{h}_i]$ share the same direction. Specifically, the middle point $\boldsymbol{m}$ and the shared direction $\boldsymbol{w}$ are as follows: $\boldsymbol{m} = (\boldsymbol{\mu}_0 + \boldsymbol{\mu}_1)/2$, and $\boldsymbol{w} = (\boldsymbol{\mu}_0 - \boldsymbol{\mu}_1)/\|\boldsymbol{\mu}_0 - \boldsymbol{\mu}_1\|_2$.*

This proposition follows from direct calculations. Given that the feature distributions of these two classes are systematic to each other (for both $\boldsymbol{x}_i$ and $\boldsymbol{h}_i$), the hyperplane that is orthogonal to $\boldsymbol{w}$ and goes through $\boldsymbol{m}$ and defines the decision boundary of the optimal linear classifier for both types of features. We denote this decision boundary as $\mathcal{P} = \left\{\boldsymbol{x} \mid \boldsymbol{w}^\top \boldsymbol{x} - \boldsymbol{w}^\top (\boldsymbol{\mu}_0 + \boldsymbol{\mu}_1)/2\right\}$. Next, to evaluate how higher-order graph convolution affects the classification performance, we compare the probability that this linear classifier misclassifies a certain node based on the features after first-order graph convolution and after the second-order graph convolution. We summarize the results in the following theorem.

**Theorem 3.1.** *Consider a graph $\mathcal{G} \sim \mathrm{cSBM}\,(\boldsymbol{\mu}_0, \boldsymbol{\mu}_1, p, q)$. For any node $i$ in this graph, the linear classifier defined by the decision boundary $\mathcal{P}$ has a lower or equivalent probability to misclassify $\boldsymbol{h}_i^2$ than $\boldsymbol{h}_i^1$ when $\deg(i) > (p+q)^2/(p-q)^2$.*

*Proof.* We only prove this for nodes from classes $c_0$ since the case for nodes from classes $c_1$ is symmetric and then the proof follows. For a node $i \in \mathcal{C}_0$, we have the follows

$$\mathbb{P}\left(\boldsymbol{h}_i^1 \text{ is mis-classified }\right) = \mathbb{P}\left(\boldsymbol{w}^\top \boldsymbol{h}_i^1 + \boldsymbol{b} \leq 0\right) \text{ for } i \in \mathcal{C}_0, \text{ and}$$
$$\mathbb{P}\left(\boldsymbol{h}_i^2 \text{ is mis-classified }\right) = \mathbb{P}\left(\boldsymbol{w}^\top \boldsymbol{h}_i^2 + \boldsymbol{b} \leq 0\right) \text{ for } i \in \mathcal{C}_0, \tag{11}$$

where $\boldsymbol{w}$ and $\boldsymbol{b} = -\boldsymbol{w}^\top (\boldsymbol{\mu}_0 + \boldsymbol{\mu}_1)/2$ is the parameters of the decision boundary $\mathcal{P}$. We have

$$\mathbb{P}\left(\boldsymbol{w}^\top \boldsymbol{h}_i^1 + \boldsymbol{b} \leq 0\right) = \mathbb{P}\left(\boldsymbol{w}^\top \sqrt{\deg(i)}\boldsymbol{h}_i^1 + \sqrt{\deg(i)}\boldsymbol{b} \leq 0\right) \text{ and}$$
$$\mathbb{P}\left(\boldsymbol{w}^\top \boldsymbol{h}_i^2 + \boldsymbol{b} \leq 0\right) = \mathbb{P}\left(\boldsymbol{w}^\top \sqrt{\deg(i)}\boldsymbol{h}_i^2 + \sqrt{\deg(i)}\boldsymbol{b} \leq 0\right). \tag{12}$$

We denote the scaled version of $\boldsymbol{h}_i^1$ and $\boldsymbol{h}_i^2$ as $\boldsymbol{h}_i' = \sqrt{\deg(i)}\boldsymbol{h}_i^1$ and $\boldsymbol{h}_i'' = \sqrt{\deg(i)}\boldsymbol{h}_i^2$ respectively. Then, $\boldsymbol{h}_i'$ and $\boldsymbol{h}_i''$ follow

$$\boldsymbol{h}_i' = \sqrt{\deg(i)}\boldsymbol{h}_i^1 \sim N\left(\frac{\sqrt{\deg(i)}\,(p\boldsymbol{\mu}_0 + q\boldsymbol{\mu}_1)}{p+q}, \boldsymbol{I}\right), \text{ for } i \in \mathcal{C}_0, \text{ and}$$
$$\boldsymbol{h}_i'' = \deg(i)\boldsymbol{h}_i^2 \sim N\left(\deg(i)(\frac{p^2\boldsymbol{\mu}_0 + 2pq\boldsymbol{\mu}_1 + q^2\boldsymbol{\mu}_1}{p+q}), \boldsymbol{I}\right), \text{ for } i \in \mathcal{C}_0. \tag{13}$$

Now, since $\boldsymbol{h}_i'$ and $\boldsymbol{h}_i''$ share the same variance, to compare the misclassification probabilities, we only need to compare the distance from their expected value to their corresponding decision boundary. Specifically, the two distances are:

$$\mathrm{dis}_{\boldsymbol{h}_i'} = \frac{\sqrt{\deg(i)}(p-q)}{(p+q)} \cdot \frac{\|\boldsymbol{\mu}_0 - \boldsymbol{\mu}_1\|_2}{2}, \text{ and } \mathrm{dis}_{\boldsymbol{h}_i''} = \frac{\deg(i)(p-q)^2}{(p+q)^2} \cdot \frac{\|\boldsymbol{\mu}_0 - \boldsymbol{\mu}_1\|_2}{2}. \tag{14}$$

The larger the distance is the smaller the misclassification probability is. Hence, when $dis_{\boldsymbol{h}_i'} < dis_{\boldsymbol{h}_i''}$, $\boldsymbol{h}_i''$ has a lower probability to be misclassified than $\boldsymbol{h}_i'$ and $\boldsymbol{x}_i$. Comparing the two distances, we conclude that when $\deg(i) > (p+q)^2/(p-q)^2$, $\boldsymbol{h}_i''$ has a lower probability to be misclassified than $\boldsymbol{h}_i'$. □

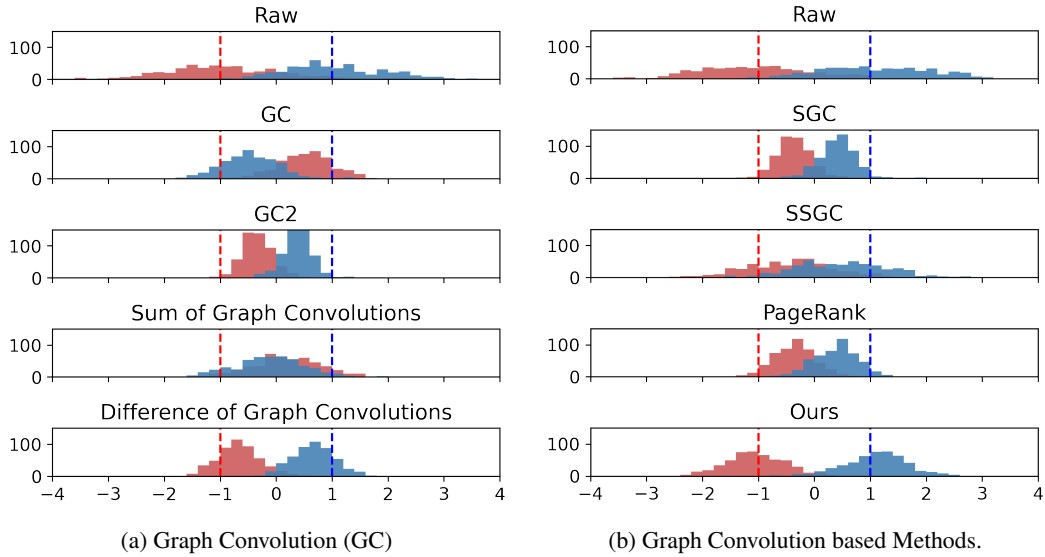

(a) Graph Convolution (GC)  (b) Graph Convolution based Methods.

Figure 2: Distribution of the feature values on a highly heterophilous synthetic graph before and after using different graph convolution based methods.

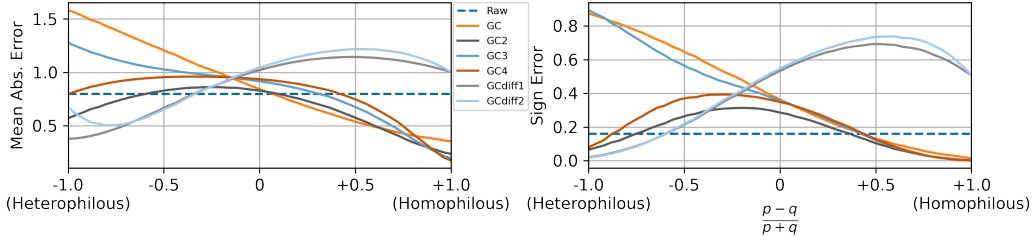

Figure 3: Results on the contextual SBM using graph convolutions with first-order (GC), second-order (GC2), third-order (GC3) and fouth-order (GC4). Two different combination of graph convolutions have been considered: the difference between first two-order (GCdiff1) and the difference between even orders and odd orders (GCdiff2).

**Theorem 3.2.** *When $p < q$ (heterophilous graphs) the presence of the parameters $w_1 < 0$ and $w_2 > 0$ allows $w_1 \boldsymbol{h}_i^1 + w_2 \boldsymbol{h}_i^2$ to have a lower probability to be misclassified than $\boldsymbol{h}_i^2$ when $\frac{w_1^2}{(1-w_2)^2} > \frac{\deg(i)(p-q)^2}{(p+q)^2}$. Please refer to the appendix for proof.*

## 4 EXPERIMENTS

### 4.1 RESULT ON CSBM SYNTHETIC

**Synthetic data** In order to test the ability of graph convolution based methods with arbitrary levels of homophily and heterophily, we use cSBMs (Deshpande et al., 2018) to generate synthetic graphs. We consider the case with two equal-size classes and take into account cSBM with n = 1000, two communities $\mathcal{C}_0$ and $\mathcal{C}_1$, feature means $\mu_0 = 1$ and $\mu_1 = -1$, and noise variance $\sigma = 1$. Then there are 500 nodes in each community, which we will refer to as "positive" and "negative," respectively. Standard normal noise is applied to the feature means of the nodes in the "positive" community, which is 1, and the "negative" community, which is $-1$. With the expected degree of all nodes set to 10 (i.e., $2(p + q)n = 10$), we create various graphs by varying the intra- and inter-community edge probabilities $p$ and $q$ from $p > q$ (highly homophilous, in that "positive" nodes are much more likely to connect to other "positive" nodes than to "negative" nodes) to $q < p$ (highly heterophilous, in that "negative" nodes. We compare our methods with three baseline models: Raw, SGC (Wu et al., 2019), S²GC (Veličković et al., 2017), PageRank (Page et al., 1999; Klicpera et al., 2018). At the sametime, we also evaluate first-order graph convolution (GC), second-order graph convolution (GC2), the sum of graph convolutions and the difference of graph convolutions.

As shown in Figure 1a, we found that some low-pass filters (SGC and APPNP) can have a positive effect on some heterophilous graphs ($\frac{p-q}{p+q} \approx -1$) with a low number of convolutions (second-order). However this phenomenon rapidly disappears as the number of convolutions increases as shown in Figure 1b, and when $K = 8$ it can be seen that all low-pass filters perform much worse than the original features on synthetic heterophilous graphs. In theoretical analysis, we prove the second-order graph convolution can provide a more discriminant features than first-order graph convolution and raw features. As shown in Figure 2a, the distribution of the feature values can qualitatively state this view. We found that in heterophilous graphs, first-order graph convolution may change the sign of the features. Thus for methods that use non-negative weights, such as PageRank, S²GC, this leads to the class centre of the features moving closer to the features' centre (global centre) as shown in Figure 2b. And the proposed method is able to keep the distance between two class centers and reduce the intra-class variance. As shown in Figure 3, we found the second-order graph convolution is better than the first-order graph convolution in graphs with different heterophilous score. The difference between second-order and first-order graph convolutions can provide a better graph convolution in heterophilous graphs while $\frac{p-q}{p+q} < -0.6$.

### 4.2 REAL WORLD BENCHMARK

We use 5 homophilous benchmark datasets available from the Pytorch Geometric library, including the citation graphs Cora, CiteSeer, PubMed (Sen et al., 2008; Yang et al., 2016) and the Amazon co-purchase graphs Computers and Photo (McAuley et al., 2015; Shchur et al., 2018). We also use 5 heterophilous benchmark datasets tested in (Pei et al., 2020), including Wikipedia graphs Chameleon and Squirrel, the Actor co-occurrence graph, and webpage graphs Texas and Cornell from WebKB. We summarize the dataset statistics and results in Table 1 and 2.

**Results on real-world datasets.** We use accuracy (the micro-F1 score) as the evaluation metric along with a 95% confidence interval. The relevant results are summarized in Table 2 . For homophilous datasets, we provide results for sparse splitting ( 2.5%/2.5%/95% splits as training/validation/test data) as same as the definition in Chien et al. (2021), which is different with the original setting used in (Kipf & Welling, 2016); (Shchur et al., 2018). For the heterophilous datasets, we adopt dense splitting ( 60%/20%/20% splits as training/validation/test data) which is used in (Pei et al., 2020). We apply our SGC, S²GC and PageRank implementations to these datasets and present the mean test accuracy over 10 randomly split data sets. We also provide a baseline on

Table 1: Statistics and results on homophilous datasets: Mean accuracy (%) $\pm$ 95% confidence interval. As expected due to design, on homophilous datasets, our method is only comparable to other graph convolution based methods because the low-pass filtering is all we need in this situation.

|  | CORA | CITESEER | PUBMED | COMPUTER | PHOTO |
|---|---|---|---|---|---|
| Nodes | 2708 | 3327 | 19717 | 13752 | 7650 |
| Edges | 5278 | 4552 | 44324 | 245861 | 119081 |
| Features | 1433 | 3703 | 500 | 767 | 745 |
| Classes | 7 | 6 | 3 | 10 | 8 |
| $H(\mathcal{G})$ | 0.825 | 0.718 | 0.792 | 0.802 | 0.849 |
| Raw | $55.09 \pm 1.81$ | $60.30 \pm 1.55$ | $77.79 \pm 0.95$ | $76.07 \pm 0.57$ | $82.97 \pm 0.58$ |
| SGC | $78.16 \pm 1.32$ | $70.18 \pm 1.00$ | $73.90 \pm 2.22$ | $87.14 \pm 0.45$ | $92.03 \pm 0.51$ |
| S$^2$GC | $78.57 \pm 1.64$ | $70.34 \pm 1.04$ | $82.89 \pm 0.46$ | $86.94 \pm 0.51$ | $92.89 \pm 0.58$ |
| PageRank | $77.65 \pm 1.70$ | $70.51 \pm 1.05$ | $75.05 \pm 1.38$ | $87.40 \pm 0.46$ | $92.95 \pm 0.57$ |
| Ours | $74.94 \pm 1.38$ | $66.86 \pm 0.86$ | $78.72 \pm 0.97$ | $86.72 \pm 0.50$ | $91.74 \pm 0.33$ |
| MLP | $50.34 \pm 0.48$ | $52.88 \pm 0.51$ | $80.57 \pm 0.12$ | $70.48 \pm 0.28$ | $78.69 \pm 0.30$ |
| GCN | $75.21 \pm 0.38$ | $67.30 \pm 0.35$ | $84.27 \pm 0.01$ | $82.52 \pm 0.32$ | $90.54 \pm 0.21$ |
| GAT | $76.70 \pm 0.42$ | $67.20 \pm 0.46$ | $83.28 \pm 0.12$ | $81.95 \pm 0.38$ | $90.09 \pm 0.27$ |
| SAGE | $70.89 \pm 0.54$ | $61.52 \pm 0.44$ | $81.30 \pm 0.10$ | $83.11 \pm 0.23$ | $90.51 \pm 0.25$ |
| JKNet | $73.22 \pm 0.64$ | $60.85 \pm 0.76$ | $82.91 \pm 0.11$ | $77.80 \pm 0.97$ | $87.70 \pm 0.70$ |
| GCN-Cheby | $71.39 \pm 0.51$ | $65.67 \pm 0.38$ | $83.83 \pm 0.12$ | $82.41 \pm 0.28$ | $90.09 \pm 0.28$ |
| GeomGCN | $20.37 \pm 1.13$ | $20.30 \pm 0.90$ | $58.20 \pm 1.23$ | NA | NA |
| APPNP | $79.41 \pm 0.38$ | $68.59 \pm 0.30$ | $85.02 \pm 0.09$ | $81.99 \pm 0.26$ | $91.11 \pm 0.26$ |
| GPRGNN | $79.51 \pm 0.36$ | $67.63 \pm 0.38$ | $85.07 \pm 0.09$ | $82.90 \pm 0.37$ | $91.93 \pm 0.26$ |

Table 2: Statistics and results on heterophilous benchmark datasets: Mean accuracy (%) $\pm$ 95% confidence interval. As expected due to design, our methods all meet or exceed the performance of raw features and are not affected by the heterophilous property like other graph convolution methods.

|  | CHAMELEON | SQUIRREL | ACTOR | TEXAS | CORNELL |
|---|---|---|---|---|---|
| Nodes | 2277 | 5201 | 7600 | 183 | 183 |
| Edges | 31421 | 198493 | 26752 | 295 | 280 |
| Features | 2325 | 2089 | 932 | 1703 | 1703 |
| Classes | 5 | 5 | 5 | 5 | 5 |
| $H(\mathcal{G})$ | 0.247 | 0.217 | 0.215 | 0.057 | 0.301 |
| Raw | $49.56 \pm 0.88$ | $34.16 \pm 0.74$ | $36.28 \pm 0.77$ | $86.49 \pm 2.88$ | $86.49 \pm 2.88$ |
| SGC | $57.70 \pm 1.62$ | $44.98 \pm 1.28$ | $30.07 \pm 0.76$ | $55.68 \pm 5.71$ | $54.32 \pm 6.41$ |
| S$^2$GC | $50.35 \pm 1.51$ | $37.77 \pm 0.78$ | $33.99 \pm 0.84$ | $71.08 \pm 4.74$ | $62.43 \pm 6.91$ |
| PageRank | $58.68 \pm 2.14$ | $42.91 \pm 0.68$ | $33.27 \pm 1.00$ | $61.89 \pm 7.03$ | $63.24 \pm 6.26$ |
| Ours | $72.28 \pm 0.90$ | $58.98 \pm 1.01$ | $36.45 \pm 0.79$ | $86.76 \pm 3.58$ | $86.49 \pm 3.08$ |
| MLP | $46.72 \pm 0.46$ | $31.28 \pm 0.27$ | $38.58 \pm 0.25$ | $92.26 \pm 0.71$ | $91.36 \pm 0.70$ |
| GCN | $60.96 \pm 0.78$ | $45.66 \pm 0.39$ | $38.02 \pm 0.23$ | $75.16 \pm 0.96$ | $66.72 \pm 1.37$ |
| GAT | $63.90 \pm 0.46$ | $42.72 \pm 0.33$ | $35.98 \pm 0.23$ | $78.87 \pm 0.86$ | $76.00 \pm 1.01$ |
| SAGE | $62.15 \pm 0.42$ | $41.26 \pm 0.26$ | $36.37 \pm 0.21$ | $79.03 \pm 1.20$ | $71.41 \pm 1.24$ |
| JKNet | $62.92 \pm 0.49$ | $44.72 \pm 0.48$ | $33.41 \pm 0.25$ | $75.53 \pm 1.16$ | $66.73 \pm 1.73$ |
| GCN-Cheby | $59.96 \pm 0.51$ | $40.67 \pm 0.31$ | $38.02 \pm 0.23$ | $86.08 \pm 0.96$ | $85.33 \pm 1.04$ |
| GeomGCN | $61.06 \pm 0.49$ | $38.28 \pm 0.27$ | $31.81 \pm 0.24$ | $58.56 \pm 1.77$ | $55.59 \pm 1.59$ |
| APPNP | $51.91 \pm 0.56$ | $34.77 \pm 0.34$ | $38.86 \pm 0.24$ | $91.18 \pm 0.70$ | $91.80 \pm 0.63$ |
| GPRGNN | $67.48 \pm 0.40$ | $49.93 \pm 0.53$ | $39.30 \pm 0.27$ | $92.92 \pm 0.61$ | $91.36 \pm 0.70$ |

the precision of logistic regression using the raw attributes without taking into account the graph convolution.

Table 1 shows that, in general, Our method cannot beat other convolution methods based on low-pass filtering designs such as SGC, S$^2$GC and PageRank on homophilous datasets. However, our approach still outperforms some classical GNNs like SAGE, JKNet, GCN-Cheby and GeomGCN. GPR-GNN achieves the state-of-the-art performance. On heterophilous datasets, our method significantly outperforms all the other graph convolution models. On Chameleon and Squirrel, we outperform other methods. It is worthy to note our approach outperform the GPR-GNN, which use the same monimial basis as our method. This is a good case to prove that the label is not necessary for heterophilous graphs. In actor, most methods cannot outperform the corresponding baseline. Except ours, all graph convolution based methods cannot outperform raw attributes with logistic regression. Similarly, only APPNP and GPRGNN can outperform raw attributes with MLP. On the Texas dataset, all methods behave similarly to those on Actor. The only difference is APPNP cannot outperform the baseline while ours and GPRGNN outperform the baseline method. Conrnell is the most challenge dataset for all methods, no one can outperform the baseline (Logistic Regression and MLP) although ours and GPRGNN can have the same performance.

## 5 CONCLUSION

From an optimalization perspective we propose an novel framework for label (or feature) propagation that is not based on Laplacian regularization. This framework extends label propagation from the least squares problem to polynomial approximation, and sheds light on graph convolution with heretophilous graphs. We show we can learn (unsupervised setting) a graph convolution that obtains better features than raw attributes. In synthetic data experiments, we show that our method has better properties on heterophilous graphs compared to existing fixed parameter graph convolutions. In real-world benchmarks, our method even outperforms some methods that use label information.

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

# A  APPENDIX

## A.1  PROOF OF THEOREM 3.2

*Proof.* We could add $w_1$ and $w_2$ into Eq. 14 and have:

$$
\begin{aligned}
\text{dis}_{comb} &= \frac{w_1\sqrt{\deg(i)}(p-q)}{(p+q)} \cdot \frac{\|\boldsymbol{\mu}_0 - \boldsymbol{\mu}_1\|_2}{2} + \frac{w_2\deg(i)(p-q)^2}{(p+q)^2} \cdot \frac{\|\boldsymbol{\mu}_0 - \boldsymbol{\mu}_1\|_2}{2} \\
&= \left( \frac{w_1\sqrt{\deg(i)}(p-q)}{(p+q)} + \frac{w_2\deg(i)(p-q)^2}{(p+q)^2} \right) \cdot \frac{\|\boldsymbol{\mu}_0 - \boldsymbol{\mu}_1\|_2}{2}.
\end{aligned}
\tag{15}
$$

We hope the $\text{dis}_{comb}$ is larger than $\text{dis}_{\boldsymbol{h}_i''}$ then we need the following inequation:

$$
\begin{aligned}
\left( \frac{w_1\sqrt{\deg(i)}(p-q)}{(p+q)} + \frac{w_2\deg(i)(p-q)^2}{(p+q)^2} \right) &> \frac{\deg(i)(p-q)^2}{(p+q)^2} \\
\frac{w_1\sqrt{\deg(i)}(p-q)}{(p+q)} &> \frac{(1-w_2)\deg(i)(p-q)^2}{(p+q)^2}.
\end{aligned}
\tag{16}
$$

We can assume $\frac{\sqrt{\deg(i)}(p-q)}{(p+q)} > 1$ and then we have:

$$
\frac{w_1^2}{(1-w_2)^2} > \frac{\deg(i)(p-q)^2}{(p+q)^2}
\tag{17}
$$

□

