# OpenReview forum: "Simple Spectral Graph Convolution from an Optimization Perspective"
_ICLR.cc/2023/Conference — Submitted to ICLR 2023_

### Official Review · Reviewer_Zwbq · 2022-10-24

**Confidence:** 4
**Correctness:** 3
**Technical Novelty And Significance:** 3
**Empirical Novelty And Significance:** 3
**Recommendation:** 6

**Clarity, Quality, Novelty And Reproducibility:**

the idea of reconstructing graph network propagated node attributes to be close to input node attributes seems intuitive

**Strength And Weaknesses:**

\+ the problem of learning graph representations without class labels is interesting and timely

\+ the results on heterophilous graph benchmarks is SOTA

\- results on homophilous benchmarks could be higher

\- writing can be improved in some parts of the paper

**Summary Of The Paper:**

This paper proposes a generalisation of the family of linear networks (SGC, SSGC, APPNP, etc.) to heterophilous graphs. In contrast to existing methods which work well with homophilous networks, this work studies an objective which can work well with heterophilous graphs. Authors demonstrate the least squares based solver based on Krylov subspaces which attempts to find a set of weights which produce a network which preserves node attributes.

**Summary Of The Review:**

Main comments:

1. In Eq. 4 and Eq. 5, do authors use class labels or attribute channels? The reviewer guesses it is attribute channels to ensure the model is an unsupervised model. This detail could be made clearer. Additionally, does Eq. 5 result in one set of weights or d set of weights assuming attributes have d channels? Could authors compare both cases?

2. Do authors ablate the impact of hyper-parameter r? Is there a sweet spot for r, or is it the best to select the largest r in each experiment?

3. Section 3.3 shows an extension of the proposed idea to optimisation over Chybyshev polynomials. Does this mean the authors could apply a simple network with weighted polynomials of degrees 1...r? If so, it would be interesting to see some result for such a problem.

4. Can authors comment on the complexity of the approach? Are Krylov subspaces used because they can find weights for consecutive diffusions $A, A^2, A^3,...$ of the linear network design or they are somehow more central to heterophily?

5. Section 3.4 provides a theoretical analysis. Are $\mu$ representing attribute centers? If p and q are inter- and intra-class probabilities of edge connectivity, does this mean p represents heterophily and q represents homophily?

6. Is there any simple strategy that could combine homophilous and heterophilious graph classification strategies?


Minor comments:

1. Dimensions of w appear to differ in Eq. 4 and Eq. 5. Do authors mean r-1 or r for the dimension of w? The same goes for Eq. 8.

2. Page 7 appears to have issues with floats (big gap between two figures).

---

### Official Review · Reviewer_VWj7 · 2022-10-25

**Confidence:** 3
**Correctness:** 3
**Technical Novelty And Significance:** 2
**Empirical Novelty And Significance:** 2
**Recommendation:** 5

**Clarity, Quality, Novelty And Reproducibility:**

The paper is well written and easy to understand. However, I didn't see a strong connection between the theoretical guarantee and proposed methods which limits the novelty of the paper. Most of the compared baseline and datasets are open source, but without further explanation on how the optimization is computed nor the code repository, it's hard to verify the reproducibility.

**Strength And Weaknesses:**

Pros:
- Training does not depends on label information hence more robust to noise and biases in training data labels.
- Solving the spectral convolution directly which can potentially avoid over-smoothing and leverage long-range interactions.
- No traditional BP process is needed.
- Extensive comparison to different GNN models and tested on various synthetic and real-world datasets.

Cons:
- Typo above equation (5), \beta is not defined.
- I think last step in eq(4) is still an approximation and not a strict equal, i.e. not an exact closed form. If it is a approximation, how do we choose r?
- The theoretical guarantee only consider up to 2nd order information, gap between 2nd order and the close-form solution is missing and only marginally support the necessity of higher order information.
- I am interested in the computational complexity needed in solving eq(5) and (8). How does the method compare to GNN based methods?
- Although the authors claims propagation on heterphily graphs should not depends on labels, the definition of H(\mathcal{G}) and cSBM is still label-dependent. Are there examples where we can see some cases it doesn’t depend on node labels?

**Summary Of The Paper:**

In this paper, the authors proposed a method to propagate node features without labels and still achieve competitive performance in both heterophilous and homophiles graphs. They reformulated the spectral convolution into a residual minimization problem and provide extensive experiments with different orders of convolution and different GNN based models.

**Summary Of The Review:**

The authors provided a different perspective on graph convolution using residual minimization and Chebyshev polynomials for approximation. However the theorems only shows the benefits of second order information, the potential complexity for higher order information in the closed form approximation is still unclear. The empirical form also ends up very similar to GPR-GNN. Without further elaboration on the necessity and complexity analysis, I am not leaning towards recommending the paper.

---

### Official Review · Reviewer_MznN · 2022-10-25

**Confidence:** 2
**Correctness:** 3
**Technical Novelty And Significance:** 3
**Empirical Novelty And Significance:** 2
**Recommendation:** 5

**Clarity, Quality, Novelty And Reproducibility:**

Overall, the paper is clearly written. The steps in the methodology description might be hard to follow, some additional intermediate explanation could be useful.

**Strength And Weaknesses:**

The paper presents the problem from an innovative perspective and proposes a link with label propagation and laplacian regularization. I believe the contribution of the paper is limited. I don't see a strong theoretical neither experimental contribution. The methodological development could benefit from more detailed explanations. For example, the claim that labels are not needed could be better empahsized.

The empirical analysis could be more comprehensive, besides the results are not fully supporting on the effectiveness of the method. In particular, one could explore more on the synthetic data side. For example, study the effect of varying parameters like the number of graphs or degree. Furthermore, especially given that it is a non deep learning based methods, it would be useful to investigate the runtime and scalability aspect.

**Summary Of The Paper:**

The paper addresses the problem of graph learning and label propagation from an optimization perspective. A theoretical framework is presented which exploit least squares to obtain a graph representation. The authors claim multiple contributions. First, it is investigated whether node labels are necessary to learn an informative representation, in the scenario of heterophilous graphs. Then, both a theoretical and experimental analysis is conducted to assess the validity of the proposed approach.

**Summary Of The Review:**

I believe the paper is valid but misses a strong contribution on the theoretical or experimental side.

---

### Official Review · Reviewer_EaVm · 2022-11-07

**Confidence:** 3
**Correctness:** 2
**Technical Novelty And Significance:** 3
**Empirical Novelty And Significance:** 2
**Recommendation:** 3

**Clarity, Quality, Novelty And Reproducibility:**

- Clarity: Writing is mostly clear, but the exposition is hard to follow. Inconsistencies in the algorithm formulation.
- Novelty: The idea itself seems novel to me.
- Reproducibility: I did not find any provided supplementary material or code.

**Strength And Weaknesses:**

## Strengths:
- The formulation in terms of the Krylov subspace is interesting and novel.
- Improvement on some heterophilous datasets (CHAMELEON and SQUIRREL)
- Theoretical analysis supports hints at some benefits of the approach on heterophilous datasets

## Weaknesses:
- The exposition is very convoluted.
- The motivation behind the core idea seems lost.
- Minor problems in the algorithm formulation.
- On most other datasets the approach underperforms.

### The exposition is very convoluted.
The related work section mentions several approaches, but the discussion provided for each of them is confusing. Although the authors have tried to connect it to the current work, these points are hard to follow. Similarly, the "Relation to GPR-GNN" section is also hard to follow, where there is new notation used but not defined.

### The motivation behind the core idea seems lost.
It seems to me that the main idea behind the approach is to substitute the closed form solution of LP into the least squares "fitting term". Why this is a good idea is not discussed at the moment. The formulation of the algorithm here seems unmotivated, and there are also minor errors, which further blurs the intuition.

I was also confused about the role of the "Polynomial Approximation with Constraints" section. The contribution of the first paragraph is tautological. Parametrizing the least squares solution in terms of the Krylov subspace is obviously equivalent to minimizing the residual with respect to all polynomials of order-r over $\mathbf{A}$. The second paragraph, on the other hand, suggests to use Chebyshev polynomials instead, but then at the end the authors say Chebyshev polynomials underperform in GNNs. No results are reported for this setting either. This makes me wonder why the authors chose to include this paragraph.

### Minor problems in the algorithm formulation.
I have found several inconsistencies in the mathematical derivation of LP and the Krylov subspace formulation. First, equation 3) seems incorrect since LP does not use the adjacency matrix $\mathbf{A}$ in the update rule, but its normalized counterpart $\mathbf{D}^{-1/2} \mathbf{A} \mathbf{D}^{-1/2}$. This follows from the Dirichlet energy term containing the division by $D_{ii}^{-1/2}$ in equation 1).

Further, in equation 4) the second equality is not true. In general, minimizing the quantity on the rightmost hand side with respect to $\mathbf{w}$ is not equivalent to minimizing the one on the middle with respect to $\alpha$. In particular, the minimum will be lower. Afterwards, the authors state that $w_i = (1 - \alpha)\alpha^i$. This contradicts the previous line, where optimization over $\mathbf{w} \in \mathbb{R}^n$ was denoted to be unconstrained. Overall, there are some inconsistencies here, which make me unsure about the actual algorithm the authors use.

### On most other datasets the approach underperforms
On the homophilous datasets, it seems like the approach is outperformed by both shallow and deep baselines. On the heterophilous datasets, it outperforms all baselines on 2 datasets, while for the other 3, it performs on par with raw logistic regression  (i.e. without connectivity information) and outperformed by deep baselines. Overall, these results are not convincing enough for me. It is not investigated where the improvements come from on the 2/5 datasets. Training time comparison against shallow/deep baselines is not reported.

**Summary Of The Paper:**

The paper proposes a novel approach for shallow graph representation through combining the idea behind label propagation with Krylov subspace methods. Label propagation is applied to the node features, and then the closed form solution is substituted into a least squares fitting term, which is then reparametrized using the Krylov subspace of order-r. Interpretation is provided as a polynomial approximation problem, and extension using Chebyshev polynomials is proposed. Theoretical analysis is provided in the context of the contextual stochastic block model in support of the approach. Experiments are carried out on synthetic data, and real-world node classification datasets.

**Summary Of The Review:**

The paper combines label propagation with optimization over Krylov subspaces. The core idea seems interesting, but the point seems lost in the convoluted exposition, technical inconsistencies in the derivation. The presentation could be improved to help better convey the idea. Experimental results only show strong performance on 2 out of 5 heterophilous datasets, while on the other 3 it performs the same as raw logistic regression, hence the feature propagation effectively having no effect. On homophilous datasets mostly weaker performance compared to baselines.

---

### Public Comment · ~Benedek_Andras_Rozemberczki1 · 2022-11-05
**Misattribution of datasets**

The paper misattributes the Chameleons and Squirrels datasets. These datasets were proposed in this ICLR submission:

https://openreview.net/forum?id=HJxiMAVtPH&referrer=%5Bthe%20profile%20of%20Carl%20Allen%5D(%2Fprofile%3Fid%3D~Carl_Allen1)

The paper cited by the authors took these datasets and used them for benchmarking. The accepted version of the ICLR submission which proposed these datasets is:
```bibtex
>@article{musae,
          author = {Rozemberczki, Benedek and Allen, Carl and Sarkar, Rik},
          title = {{Multi-Scale Attributed Node Embedding}},
          journal = {Journal of Complex Networks},
          volume = {9},
          number = {2},
          year = {2021},
}
```

---

> ### Author Response · Authors · 2022-11-19
> **Thank you.**
>
> Thank you. We will of course cite the above work accordingly.

---

### Public Comment · ~Sitao_Luan1 · 2022-11-14
**Relevant Work**

Thank the authors for having this interesting paper which leverages Krylov subspace for graph representation learning. I would like to highlight one relevant work [1] that proposes truncated Krylov and snowball networks for deep GNNs. If you are interested in heterophily problem and would like to know how snowball GNNs can perform well on heterophilic graphs, we want to introduce our recent work [2]. Good luck to your rebuttal.

[1] Luan S, Zhao M, Chang X W, et al. Break the ceiling: Stronger multi-scale deep graph convolutional networks[J]. Advances in neural information processing systems, 2019, 32.

[2] Luan S, Hua C, Lu Q, et al. Revisiting Heterophily For Graph Neural Networks[J]. NeurIPS 2022. arXiv:2210.07606, 2022.

---

> ### Author Response · Authors · 2022-11-19
> **Thank you.**
>
> Thank you for pointing out interesting works.
>
> We of course will cite them accordingly.
>
> Kindly notice that the above two works are **supervised graph networks** (as far as we understand them correctly) while **we work with the family of unsupervised linear graph networks** which compete with **unsupervised and self-supervised graph representations** rather than fully-supervised networks (completely different working regimes in terms of speed etc.)
>
> Nonetheless, the connection of working on heterophilic graphs is relevant as prior work. Thank you again for pointing out these nice works.
>
> Best regards,
> \
> Authors

---

### Decision · Program_Chairs · 2023-01-20

**Decision:**

Reject

**Justification For Why Not Higher Score:**

The paper is clearly below bar for publication at ICLR

**Justification For Why Not Lower Score:**

NA

**Metareview: Summary, Strengths And Weaknesses:**

The paper proposes a graph representation by combining label propagation with Krylov subspace. Label propagation is applied to the node features, and then the closed form solution is substituted into a least squares fitting term, which is then reparametrized using the Krylov subspace.  sOME Theoretical analysis is provided. Experiments are carried out on synthetic setting, and real-world node classification datasets.

The formulation is interesting and novel. The method provide improvement on CHAMELEON and SQUIRREL dataset

However, the method does not perform well on other datasets. The claim made by authors that the method is not using all the information and hence is unlikely to outperform other strong baselines is generally not a valid support for the arguments. The performance in many case is almost as good as vanilla logistic regression. As a result, the real value of doing representation learning does not stands out and hence the empirical evaluation on real datasets fails to support the need for such methods in practice.  The empirical results are inconclusive.

**Summary Of Ac-Reviewer Meeting:**

Reviewers in generally found the manuscript to be below bar for publication at ICLR.  The lack of convincing experiments in the paper was the primary reason for the lack of excitement. Rebuttal brought in substantial new information and experiments resulting in major overhaul of the paper. However, the substantial changes require the paper to go through a full review cycle.